# ScandEval: A Benchmark for Scandinavian Natural Language Processing

**Dan Saattrup Nielsen**
The Alexandra Institute
Copenhagen, Denmark
`dan.nielsen@alexandra.dk`

## Abstract

This paper introduces a Scandinavian benchmarking platform, `ScandEval`, which can benchmark any pretrained model on four different tasks in the Scandinavian languages. The datasets used in two of the tasks, linguistic acceptability and question answering, are new. We develop and release a Python package and command-line interface, `scandeval`, which can benchmark any model that has been uploaded to the Hugging Face Hub, with reproducible results. Using this package, we benchmark more than 100 Scandinavian or multilingual models and present the results of these in an interactive online leaderboard[1], as well as provide an analysis of the results. The analysis shows that there is substantial cross-lingual transfer among the Mainland Scandinavian languages (Danish, Swedish and Norwegian), with limited cross-lingual transfer between the group of Mainland Scandinavian languages and the group of Insular Scandinavian languages (Icelandic and Faroese). The benchmarking results also show that the investment in language technology in Norway, Sweden and Denmark has led to language models that outperform massively multilingual models such as XLM-RoBERTa and mDe-BERTaV3. We release the source code for both the package[2] and leaderboard[3].

## 1 Introduction

In recent years, there has been a significant increase in the number of monolingual language models in the Scandinavian languages (Møller-høj, 2020; Højmark-Bertelsen, 2021; Sarnikowski, 2021; Enevoldsen et al., 2021; Abdaoui et al., 2020; Kummervold et al., 2021; Malmsten et al., 2020; Snæbjarnarson et al., 2023), to the extent that it becomes difficult both for the practioner to choose the best model for the task at hand, as well as for language researchers to ensure that their research efforts are indeed improving upon past work.

Aside from the increasing number of models, Sahlgren et al. (2021) also emphasises that a joint Scandinavian language model is probably a better strategy for the Scandinavian countries, considering the similarity of their languages and culture. Indeed, Faarlund (2019) even argues that the Danish, Norwegian and Swedish languages are so similar that they should be considered a single language.

The languages included in the term "Scandinavian" is debatable (oxf, 2021). Following the distinction between *Mainland Scandinavian* (Danish, Swedish and Norwegian) and *Insular Scandinavian* (Icelandic and Faroese) (Haugen, 1976; Faarlund, 2019), a distinction based on mutual intelligibility and syntactical structure, we focus in this work on the Mainland Scandinavian languages, while still allowing support for the Insular Scandinavian languages. Aside from being a standard distinction, our choice is also based on experiments on the cross-lingual transfer between these two groups, which we present in Section 3.3. We will here use the term "Scandinavian" to mean the collection of all five languages, and use the Mainland/Insular distinction when applicable.

To help facilitate progress in both improving upon the monolingual Scandinavian models as well as the multilingual, we present `ScandEval`, a benchmark of Scandinavian models, along with a Python package and Command-Line Interface (CLI), and an associated online leaderboard. This leaderboard contains the results of language models benchmarked on datasets within the Mainland

---

[1] `https://scandeval.github.io`
[2] `https://github.com/saattrupdan/ScandEval`
[3] `https://github.com/ScandEval/scandeval.github.io`

Scandinavian languages, as described in Section 4.

Recent studies (Khanuja et al., 2021; Pires et al., 2019; Lauscher et al., 2020) have shown that multilingual models can outperform monolingual models when the languages are sufficiently similar, and also that they are worse than the monolingual models when the languages are too dissimilar. This shows that the Scandinavian languages could have something to gain by creating "local multilingual" models, rather than using the massively multilingual models such as `XLM-RoBERTa` (Conneau et al., 2020). Based on this, we test the following hypotheses:

- **Hypothesis 1:** There is a substantial cross-lingual transfer within the Mainland Scandinavian languages.

- **Hypothesis 2:** There is no notable cross-lingual transfer between the group of Mainland Scandinavian languages and the group of Insular Scandinavian languages.

To the best of our knowledge, this is the first benchmarking tool for any of the Scandinavian languages, as well as the first online leaderboard containing scores from such a tool. Our contributions are the following:

1. We construct a new question answering dataset for the Mainland Scandinavian languages, dubbed `ScandiQA`.

2. We construct a new linguistic acceptability dataset for all the Scandinavian languages, dubbed `ScaLA`.

3. We develop a Python package and CLI, `scandeval`, which allows reproducible benchmarking of language models on Scandinavian language datasets.

4. We uniformise all the datasets used in the benchmark, to enable consistent evaluation across languages and datasets. These uniformised datasets are also available on the Hugging Face Hub[4].

5. We benchmark all the Scandinavian and a selection of the multilingual language models on the Hugging Face Hub[5] on the Mainland Scandinavian datasets in the benchmark, and present all the scores in an online leaderboard.

[4]https://huggingface.co/ScandEval
[5]https://hf.co

## 2 Related Work

There has been a number of (non-English) NLU benchmarks published in recent years (Wang et al., 2018; Sarlin et al., 2020; Rybak et al., 2020; Ham et al., 2020a; Shavrina et al., 2020; Wilie et al., 2020; Xiang et al., 2021; Koto et al., 2020; Safaya et al., 2022; Augustyniak et al.; Khashabi et al., 2020; Ham et al., 2020b; Xu et al., 2020; Dumitrescu et al., 2021), with whom we share the same goal of advancing the state of NLP in our respective languages. Within the Scandinavian languages specifically, the SuperLim benchmark (Adesam et al., 2020) is a Swedish NLU Benchmark featuring several difficult tasks. Most of the datasets in the SuperLim benchmark only contain a test set, however.

The `XGLUE` (Liang et al., 2020) dataset is another multilingual NLU benchmark. That dataset is different from `ScandEval` in that all the training data in `XGLUE` is in English, and that the majority of the test sets are not available in any of the Scandinavian languages.

Isbister and Sahlgren (2020) present a Swedish similarity benchmark, achieved through machine translating the `STS-B` dataset from the GLUE benchmark (Wang et al., 2018). Aside from only dealing with a single task and a single language, the quality of the dataset is worse than a gold-standard corpus as a result of the translation, as the authors also point out.

## 3 Methodology

This section describes our benchmarking methodology in detail, including both the setup of the datasets, the evaluation procedure and the scoring of the models. We also describe how we conduct the cross-lingual transfer experiments.

### 3.1 Finetuning Setup

When finetuning, we enforce a learning rate of $2 \cdot 10^{-5}$ with 100 warmup steps, and a batch size of 32. If there is not enough GPU memory to finetune the model with this batch size, we halve it and double the amount of gradient accumulation, resulting in the same effective batch size. This is repeated until the batches can fit in memory.

We impose a linear learning rate schedule with intercept after $10,000$ training steps (with a training step consisting of 32 samples), and we adopt early stopping (Plaut et al., 1986) to stop the training procedure if the validation loss has not de-

creased for 90 training steps. We use the `AdamW` optimiser (Loshchilov and Hutter, 2018) with first momentum $\beta_1 = 0.9$ and second momentum $\beta_2 = 0.999$, and we optimise the cross-entropy loss throughout all tasks. Further, random seeds are fixed throughout, to ensure reproducibility.

The finetuning itself uses the `transformers` package (Wolf et al., 2020). For the named entity recognition task we use the `AutoModelForTokenClassification` class, which linearly projects the embedding from the language model encoder for each token into the entity logits for that token. For the classification tasks we use the `AutoModelForSequenceClassification` class, which linearly projects the embedding from the language model encoder to each document into the class logits for that document. Lastly, for the question answering task we use the `AutoModelForQuestionAnswering` class, which linearly projects the embedding from the language model encoder for each token, into the logits of the start and end positions of the answer for that token.

### 3.2 Bootstrapping Evaluation

For each model and dataset, we repeat the following procedure 10 times, which generates a score for each model and dataset combination: (a) Fix a random seed unique to the given iteration; (b) Finetune the model on the training set; (c) Evaluate the model on a bootstrapped (i.e., sampling with replacement) version of the test set. The evaluation score is then the mean $\mu$ of these scores, along with a 95% confidence interval $I_{10}$, computed as

$$I_N := \mu \pm \frac{1.96}{N-1} \sum_{i=1}^{N} \text{score}_i. \quad (1)$$

The combination of varying the random seeds as well as using bootstrapped test datasets ensures that we capture the noise coming from both the random initialisation of the added layers to the model as well as the noise in the test set, resulting in a more reliable confidence interval of the true mean for each model and dataset combination.

To aggregate these scores across all datasets, we firstly compute the *language-specific task scores* for each (model, language, task) triple, which is the mean of the scores of the model on the tasks of the language.[6] From these language-specific

scores we next compute the *language score* for each (model, language) pair as the mean of the language-specific task scores across all the tasks. A final `ScandEval` score is computed as the average of the language scores, to emphasise the training of Scandinavian models rather than monolingual ones.

### 3.3 Cross-lingual Transfer

To test Hypothesis 1 and 2, stated in Section 1, we introduce a way to measure the "joint cross-lingual transfer" of a group of languages, by which we mean an aggregate of the cross-lingual transfer between any two languages in the group.

To do this, we first introduce a control group of non-Scandinavian languages: English, German, Dutch, Finnish, Russian and Arabic. By considering the combined set of languages in the control group and the Scandinavian languages, we aim to find the best split of these languages into two groups: a ScandEval benchmark group and a non-benchmark group. The "goodness" of a split is measured by benchmarking a "representative" model from each language on datasets in each of the benchmark languages and measuring the quality of the two-cluster clustering of these benchmarking values.

As an example, if Danish and Swedish constitute the benchmark group and the rest of the languages are in the non-benchmark group, we would benchmark the representative models from each language on the Danish and Swedish part of ScandEval, and then compute the F-statistic of the clustering $\{\{\text{da}, \text{sv}\}, \{\text{no}, \text{is}, \text{fo}, \ldots\}\}$ with these benchmarking values, computed as the ratio of the between-group variance to the within-group variance.[7] We can then compare this F-statistic to the F-statistic of the clustering where the benchmark group consists of Danish and Norwegian, for instance.

As for picking a representative model for each language, we found pretrained language models of roughly the same size on the Hugging Face Hub, each of which has been pretrained on solely monolingual data. We note that no Faroese language model exists, so for that language we do not include any model but still include Faroese benchmarking

---

[6]This mean is only non-trivial for the Norwegian language

for the named entity recognition task and the linguistic acceptability task, as these tasks are available in both Norwegian Bokmål and Norwegian Nynorsk.

[7]Technically speaking, we get an F-statistic for each language in the benchmarking group, but we just use the mean of these F-statistics.

datasets when Faroese is part of the benchmarking group. See the full list of models in the appendix.

We can then restate our first hypothesis as the mainland Scandinavian languages are all in the best-performing benchmark group, and our second hypothesis as the Insular Scandinavian languages are not in the best-performing benchmark group.

### 3.4 Uniform Benchmarking Datasets

As we are interested in comparing the performance of the models across languages, we ensure that all the datasets used in the benchmark are of the same format and the same size.

We aimed to choose a training data size that would be a balance between being able to differentiate between the models and being able to benchmark the models in a reasonable amount of time. We benchmarked the same models as in Section 3.3 on truncations of named entity recognition datasets, sentiment classification datasets and linguistic acceptability datasets. Based on these results we qualitatively found that using 1,024 training samples allowed for both differentiation between the models and being able to benchmark the models in a reasonable amount of time. Figures 1 and 2 show the trade-off between differentiation and benchmarking speed, covering the `AngryTweets` dataset (Pauli et al., 2021). The remaining plots for the other datasets can be found in the appendix.

Another benefit of using a small training dataset is that it emphasises the importance of the pre-trained weights of the models, rather than the fine-tuning process. Further, we wanted the test dataset to be as large as possible, to ensure more robust evaluations of the models, which led to the choice of 2,048 test samples based on the number of available samples in the smallest dataset. Lastly, the validation set was chosen to be 256 samples, to allow for a reasonable evaluation during training, while not being too time-consuming. All of these datasets with their splits are available on the Hugging Face Hub.

## 4 ScandEval Tasks

To properly evaluate the performance of a pre-trained model, we ideally need to evaluate it on many diverse tasks. Unfortunately, the Scandinavian languages do not have many openly available datasets for many downstream tasks.

To address this, we construct two new Scandinavian datasets, `ScaLA` and `ScandiQA`, being *Lin-*

*guistic Acceptability* (`LA`) and *Question Answering* (`QA`) datasets, respectively. These new tasks are supplemented by existing benchmarking datasets within *Named Entity Recognition* (`NER`) and *Sentiment Classification* (`SENT`). Aside from downstream performance of these tasks, we also benchmark the inference speed of each model. We describe all of these in more detail in the subsections below.

### 4.1 Named Entity Recognition

For the NER task we use the four classes used in CONLL (Tjong Kim Sang and De Meulder, 2003): `PER`, `LOC`, `ORG` and `MISC`, corresponding to person names, locations, organisations and miscellaneous entities.

Since this is a token classification task and that the language models usually use different tokenisers, we have to ensure a uniform treatment of these as well. We tokenise the documents using the pre-trained tokeniser associated to the model that we are benchmarking, and to ensure consistency of the evaluation we replace all but the first token in each word with the empty entity `O`. For instance, if the word "København" with the `LOC` tag is tokenised as ["Køben", "havn"], then we would assign the labels `LOC` and `O` to these tokens. This ensures that we maintain the same number of (non-empty) labels per document.

In terms of evaluation metrics, we use the micro-average F1-score, which is standard for NER. We also report a *no-misc score*, which is the micro-average F1-score after we replace the `MISC` class in the predictions and labels with the "empty label" `O`. This *no-misc score* is not used in any of the aggregated scores and is purely used for comparison purposes on the individual datasets.

For Danish we use the `DaNE` dataset (Hvingelby et al., 2020), being a NER tagged version of the Danish Dependency Treebank (Kromann and Lynge, 2004). `DaNE` is already in the CONLL format, so we perform no preprocessing on the data.

For Norwegian we use the Bokmål and Nynorsk `NorNE` datasets (Jørgensen et al., 2020), also being NER tagged versions of the Norwegian Dependency Treebanks (Øvrelid and Hohle, 2016). Aside from the `PER`, `LOC`, `ORG` and `MISC` tags, these also include `GPE_LOC`, `GPE_ORG`, `PROD`, `DRV` and `EVT` tags. We convert these to `LOC`, `ORG`, `MISC`, `MISC` and `MISC`, respectively.

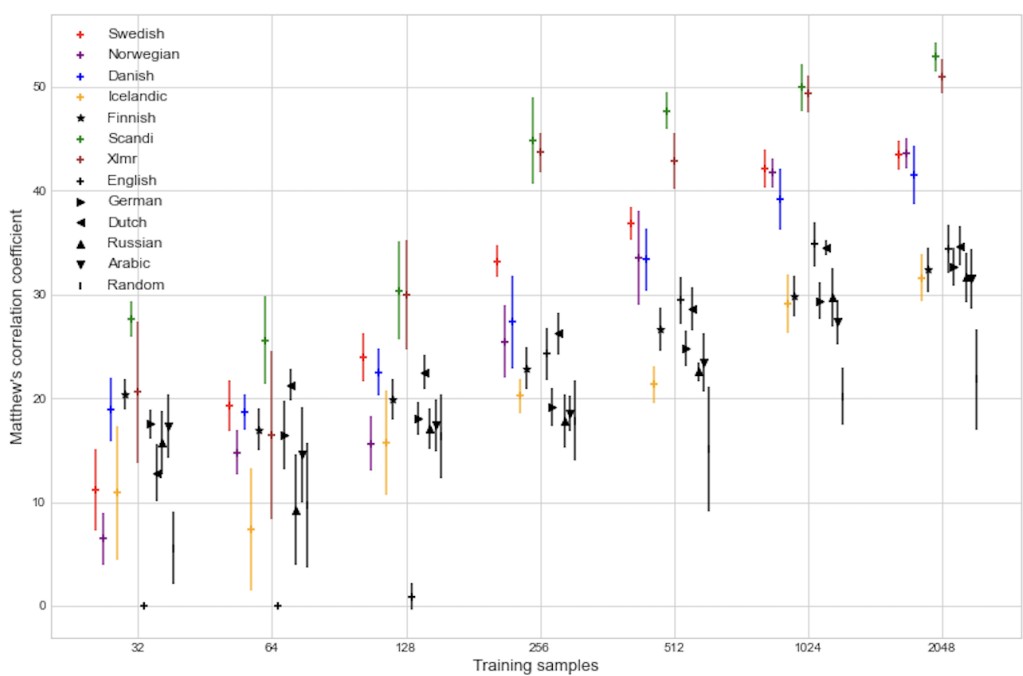

Figure 1: Plot showing the performance of different models on the `AngryTweets` dataset with varying number of training samples.

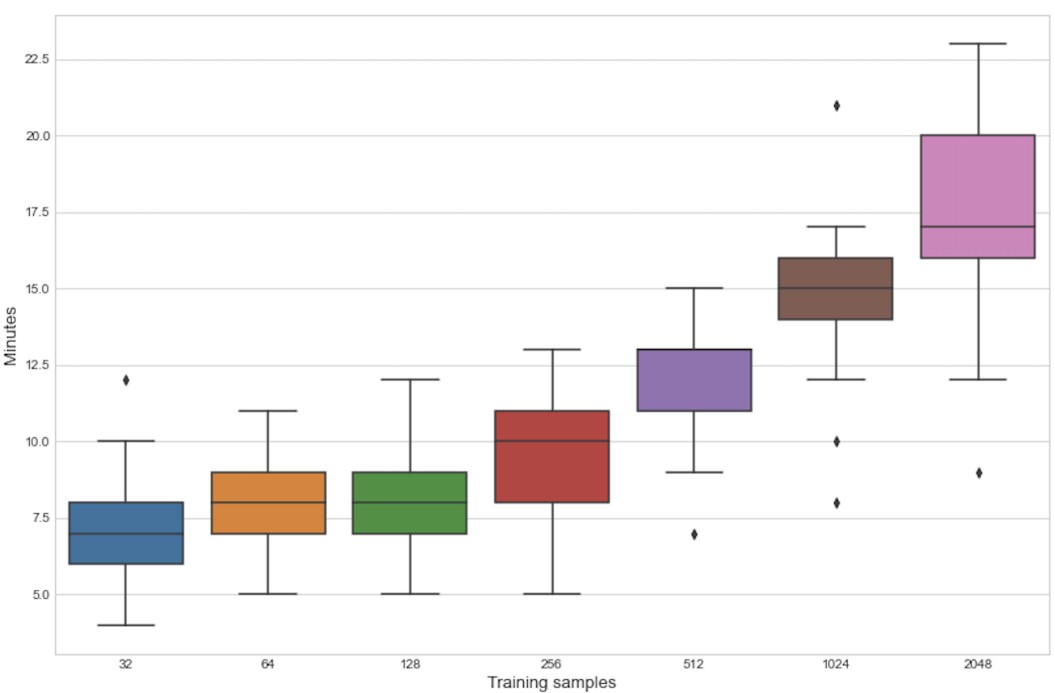

Figure 2: Boxplot showing the training time of the models on the `AngryTweets` dataset with varying number of training samples.

Lastly, Swedish does not have a NER tagged version of the corresponding dependency treebank, but they instead have the `SUC3` dataset, a NER-enriched version of the *Stockholm-Umeå Corpus*

(Gustafson-Capková and Hartmann, 2006). This dataset does not follow the CONLL format and is instead released in the XML format, with the `<name>` XML tags containing the NER tags for

the words they span over [8]. This dataset contains the NER tags `animal`, `event`, `inst`, `myth`, `other`, `person`, `place`, `product` and `work`. These were converted to `MISC`, `MISC`, `ORG`, `MISC`, `MISC`, `PER`, `LOC`, `MISC` and `MISC`, respectively.

## 4.2 Sentiment Classification

We treat the sentiment classification task as a three-class classification task, with the classes `positive`, `neutral` and `negative`. Evaluation of the models is done using Matthew's Correlation Coefficient (Matthews, 1975) as the primary metric as well as reporting the macro-average F1-score as a secondary metric. We choose to use Matthew's Correlation Coefficient as the primary metric as it has been shown to be more reliable than the macro-average F1-score (Chicco and Jurman, 2020), while also being the standard metric used in the `GLUE` (Wang et al., 2018) and `SuperGLUE` (Sarlin et al., 2020) benchmarks.

For Danish we use the sentiment classification dataset `AngryTweets` (Pauli et al., 2021), which contains crowdsourced annotations of Danish tweets. To comply with Twitter's Terms of Use we have fully anonymised the tweets by replacing all user mentions with `@USER` and all links by `[LINK]`, as well as shuffling the tweets.

For Norwegian we included the sentiment classification dataset `NoReC` (Norwegian Review Corpus) (Velldal et al., 2018), which are based on scraped reviews from Norwegian websites.

Lastly, for Swedish we use the sentiment classification dataset presented in Svensson (2017), which is based on reviews from the Swedish websites `www.reco.se` and `se.trustpilot.com`. In analogy with `NoReC` we dub this dataset the Swedish Review Corpus (`SweReC`).

## 4.3 Linguistic Acceptability

Based on the inclusion of the CoLA (Corpus of Linguistic Acceptability) dataset (Warstadt et al., 2019) in the GLUE benchmark (Wang et al., 2018), we construct new linguistic acceptability datasets for the Scandinavian languages. This task is often framed as a binary classification task, where the model is tasked with predicting whether a given sentence is grammatically correct or not.

We dub our new datasets Scandinavian Linguistic Acceptability (`ScaLA`), which we release for Danish, Norwegian Bokmål, Norwegian Nynorsk, Swedish, Icelandic and Faroese. Each of these datasets consist of 1,024 training samples, 256 validation samples and 2,048 test samples, in accordance with Section 3.4. The `ScaLA` datasets are based on the Danish, Norwegian, Swedish, Icelandic and Faroese versions of the Universal Dependencies datasets (Kromann and Lynge, 2004; Øvrelid and Hohle, 2016; Nivre et al., 2006; Rögnvaldsson et al., 2012; Jónsdóttir and Ingason, 2020; Arnardóttir et al., 2020).

Firstly, we assume that the documents in the Universal Dependencies datasets are grammatically correct, an assumption we have been able to verify for the Danish part, by manually inspecting a random sample of the documents. We create negative examples by *either* removing a single word or swapping two consecutive words, where only one such "corruption" is applied to each negative sample.

Naively corrupting the documents in this way does not always lead to grammatically incorrect samples, however. For instance, removing the word "rød" (red) from the sentence "Den røde bil er stor" (The red car is big) does not lead to an incorrect sentence "Den bil er stor" (The car is big).

In order to ensure that the resulting sentence is indeed grammatically correct, we enforce restrictions on the words that can be removed or swapped. We have gone for a conservative approach, where we have systematically checked corruptions of words with a given part-of-speech tag, and only allow corruptions that were always grammatically correct in our tests. This led us to the following restrictions:

1. We do not remove adjectives, adverbs, punctuation, determiners or numbers, as the resulting sentence will still be grammatically correct in most cases.

2. We do not remove nouns or proper nouns if they have another noun or proper noun as neighbour, as again that usually does not make the sentence incorrect either.

3. When swapping two neighbouring words, we require them to have different POS tags.

4. We do not swap punctuation or symbols.

5. If we swap the first word then we ensure that the swapped words have correct casing.

---

[8]The `<ne>` XML tags are also NER tags, but these have been automatically produced by `SpaCy` (Honnibal et al., 2020) models and are thus not gold standard.

We are able to enforce these restrictions as we have gold-standard POS tokens available for these datasets.

## 4.4 Question Answering

We also construct new question answering datasets for the Mainland Scandinavian languages, as we are not aware of any existing datasets for these languages. We dub these datasets `ScandiQA`, which we release for each of the Mainland Scandinavian languages.

These datasets are based on the `MKQA` dataset (Longpre et al., 2021), which is based on the *Natural Questions* (`NQ`) dataset (Kwiatkowski et al., 2019). The `NQ` dataset contains questions inputted to Google's search engine, associated with the HTML page of the search result. In many cases these questions have an answer associated with it (a so-called *short answer*) which appears in the HTML, and in some cases they also have the paragraph in which the short answer appears (a so-called *long answer*).

The `MKQA` dataset contains human translations of 10,000 questions and short answers into 26 languages, including Danish, Norwegian and Swedish. Aside from adding these translations, the `MKQA` dataset also corrects many mistakes in the original `NQ` dataset by including answers not present in the original dataset, or by correcting the short answers chosen in the original dataset.

The main thing missing from the `MKQA` dataset is the context paragraph, which is what we add to the dataset as follows. For each `MKQA` sample, we first locate the corresponding sample in the `NQ` dataset. If that sample has a long answer then we use that as the initial (English) context. Otherwise, if neither the `NQ` dataset nor the `MKQA` dataset has an answer registered, then we use the paragraph in the HTML with the largest cosine similarity to the question, where we embed the documents using the Sentence Transformer (Reimers and Gurevych, 2019) model `all-mpnet-base-v2`.[9]

In the last case, where there is no long answer for the sample in `NQ` but there *is* an answer in `MKQA`, we want to identify the paragraph in the HTML containing the `MKQA` answer. Unfortunately, the `MKQA` answers do no appear verbatim in the HTML (for instance, all dates are standardised to the `YYYY-MM-DD` format). We thus start by forming a list of *answer candidates* based on the `MKQA` answer, which includes most of the ways dates and numerals are written in English. We then locate the paragraph containing any of the answer candidates and which has the largest cosine similarity to the question, where we embed the documents as described above.

The above procedure thus results in an English context paragraph containing the answer. We next translate this context paragraph to Danish and Swedish using the DeepL translation API[10]. As DeepL did not support Norwegian when we conducted this experiment, we translated the context paragraph to Norwegian using the Google Translation API[11] instead. With the contexts translated, we next extract all the answer candidates for the translated context relevant to the given Mainland Scandinavian language, and change the answer to the answer candidate appearing in the translated context. If no answer candidate appears in the translated context then we discard the sample.

The `MKQA` dataset also contains samples with *no* answer, and we include these samples in the `ScandiQA` dataset as well. For these samples, we simply use the translated context paragraphs as described above. The final dataset contains 7,810 Danish samples, 7,798 Swedish samples and 7,813 Norwegian samples. We release this dataset separately[12], as well as build a ScandEval version of it with the same train/dev/test size as the other ScandEval datasets. In the ScandEval version (with 1,024/256/2,048 train/val/test samples as stated in Section 3.4) we only include samples that contain an answer, as otherwise we found the 1,024 dataset size to be too small for this task.

We note that since this dataset is a translated version of a dataset originally written in English, it is not a perfect representation of the Mainland Scandinavian languages, as many of the questions and answers are concerned with topics specific to the USA. This might mean that pretrained multilingual models might have an advantage over monolingual models, but we leave this question for future work.

---

[9] https://huggingface.co/sentence-transformers/all-mpnet-base-v2

[10] https://www.deepl.com/pro-api

[11] https://cloud.google.com/translate/

[12] This can be found at https://huggingface.co/datasets/alexandrainst/scandi-qa and the source code is available at https://github.com/alexandrainst/ScandiQA.

## 4.5 Inference Speed

Aside from the predictive performance of the models we also benchmarked the inference speed of the finetuned models using the `pyinfer` package (Pierse, 2020), and report the mean number of inferences per second. This is done by recording the mean inference time of running a document with 2,600 characters[13] through the model one hundred times, and repeating that process 10 times. We also compute the confidence interval as described in Section 3.2. These have all been computed using an AMD Ryzen Threadripper 1920X 12-Core CPU.

## 5 Benchmarking Package and CLI

To enable every language researcher to benchmark their language models in a reproducible and consistent manner, we have developed a Python package called `scandeval`, which can benchmark any pretrained language model available on the Hugging Face Hub.

The `scandeval` package is implemented as both a CLI and a Python package, which enables ease of use as both a stand-alone benchmarking tool as well as enabling integration with other Python scripts. The package follows a very *opinionated* approach to benchmarking, meaning that very few parameters can be changed. This is a deliberate design decision to enable consistent benchmarking of all models. The package follows the hyperparameter choices described in Section 3.1. See more in the `scandeval` documentation.

## 6 Experiments

Using the `scandeval` package we have benchmarked more than 100 pretrained models in the Scandinavian languages which were available on the Hugging Face Hub. Aside from these models we also included several multilingual models to enable a fair comparison. Lastly, to enable better interpretability of the results, we also benchmark a randomly initialised XLM-RoBERTa-base model (Conneau et al., 2020) and an ELECTRA-small model (Clark et al., 2019) on the datasets, which will make it more transparent how much "external knowledge" the pretrained models are able to utilise in their predictions. Benchmarking all these models approximately required 1000 GPU hours

on a GeForce RTX 2080 Ti GPU, which emitted approximately 40 kg of $CO_2$ equivalents[14].

## 6.1 Benchmarking Results

We have presented all of the benchmarked results along with their associated confidence intervals in an online leaderboard. These scores have been computed as described in Section 3, and the top-5 performing models for each language, as well as overall, can be found in Table 1.

We see from Table 1 that NB-BERT-large[15] (Kummervold et al., 2021) is the best performing model in Norwegian as well as overall, DFM-encoder-large-v1[16] being the best Danish model, and KB-BERT-large[17] (Malmsten et al., 2020) having the best performance in Swedish.

The massively multilingual models in the top 5 scores are RemBERT (Chung et al., 2020) and mDeBERTaV3 (He et al., 2021). The remaining models in the top 5 are NB-RoBERTa-base-scandi[18], DanskBERT (Snæbjarnarson et al., 2023), NB-BERT-base (Kummervold et al., 2021), Nor-BERT2 (Kutuzov et al., 2021), KB-BERT-base (Malmsten et al., 2020) and AI-Nordics-BERT-large[19].

## 6.2 Cross-lingual Transfer

This experiment investigated the cross-lingual transfer capabilities of the Scandinavian models, and tested our two hypotheses from Section 1. This used the methodology described in Section 3.3. For the Insular Scandinavian languages, the tasks included here are the Icelandic and Faroese versions of the `ScaLA` dataset, the Icelandic `NER` dataset `MIM-GOLD-NER` (Ingólfsdóttir et al., 2020) and the Faroese part of the `NER` dataset `WikiANN` (Rahimi et al., 2019). The resulting benchmark results can be found in Table 2 and all the raw scores can be found in the appendix. The results affirm our two hypotheses, as we see that the group

---

[13]The document is "This is a dummy document. ", repeated 100 times.

[14]With a power usage of 250 W/h (Techpowerup.com) and a carbon efficiency of 0.16 kg/kWh in Denmark (Ritchie et al., 2022).

[15]https://huggingface.co/NbAiLab/nb-bert-large

[16]https://huggingface.co/chcaa/dfm-encoder-large-v1

[17]https://huggingface.co/KBLab/megatron-bert-large-swedish-cased-165k

[18]https://huggingface.co/NbAiLab/nb-roberta-base-scandi

[19]https://huggingface.co/AI-Nordics/bert-large-swedish-cased

| Rank | Overall | Danish | Norwegian | Swedish |
|------|---------|--------|-----------|---------|
| 1 | NB-BERT-large | DFM-encoder-large-v1 | NB-BERT-large | KB-BERT-large |
| 2 | DFM-encoder-large-v1 | NB-BERT-large | NB-BERT-base | NB-BERT-large |
| 3 | RemBERT | DanskBERT | NB-RoBERTa-base-scandi | KB-BERT-base |
| 4 | mDeBERTaV3-base | RemBERT | NorBERT2 | AI-Nordics-BERT-large |
| 5 | NB-RoBERTa-base-scandi | mDeBERTaV3-base | mDeBERTaV3 | RemBERT |

Table 1: The five best performing pretrained models in the Mainland Scandinavian language categories.

of languages with the largest F-statistic is the group of Mainland Scandinavian languages.

| Benchmark group | F-statistic | Benchmark group | F-statistic |
|-----------------|-------------|-----------------|-------------|
| da, no | 16.81 | da, sv, is | 4.36 |
| da, sv | 15.48 | da, sv, fo | 10.72 |
| da, is | 4.76 | da, is, fo | 5.48 |
| da, fo | 7.29 | no, sv, is | 3.11 |
| no, sv | 8.14 | no, sv, fo | 5.57 |
| no, is | 3.73 | no, is, fo | 3.64 |
| no, fo | 2.70 | sv, is, fo | 4.26 |
| sv, is | 4.48 | da, no, sv, is | 6.97 |
| sv, fo | 7.59 | da, no, sv, fo | 25.40 |
| is, fo | 21.84 | da, no, is, fo | 4.97 |
| **da, no, sv** | **33.34** | da, sv, is, fo | 5.21 |
| da, no, is | 4.27 | no, sv, is, fo | 3.38 |
| da, no, fo | 11.56 | da, no, sv, is, fo | 7.53 |

Table 2: F-statistics showing the cross-lingual transfer between the Scandinavian language models. Here da is Danish, no is Norwegian, sv is Swedish, is is Icelandic and fo is Faroese.

## 7   Discussion

We note that the benchmarking results presented in Section 6.1 show that the efforts of the National Libraries in Norway and Sweden, as well as the Danish Foundation Models project in Denmark, have paid off, in the sense that their models NB-BERT-large (Kummervold et al., 2021), KB-BERT-large (Malmsten et al., 2020) and DFM-encoder-large-v1 are outperforming the multilingual models.

This seems to indicate that investing in language technologies at a large language-specific level can be worthwhile. We also see from the same table that the Norwegian model is within the top two best models in Danish, Norwegian and Swedish, indicating a potentially large amount of language transfer, supported by the cross-lingual transfer experiment in Section 6.2. This indicates that a joint Mainland Scandinavian approach could improve the results of the current monolingual models within the Mainland Scandinavian languages.

## 8   Conclusion

In this paper we have presented a benchmarking framework for the Scandinavian languages, together with a Python package and CLI,

scandeval, which can be used to benchmark any model available on the Hugging Face Hub. The benchmark features four tasks: named entity recognition, sentiment classification, linguistic acceptability and question answering. We have also released two new datasets, ScaLA and ScandiQA, which constitute the linguistic acceptability and question answering tasks, respectively. We have benchmarked more than 100 models on the Mainland Scandinavian datasets in the benchmark and presented these results in an online leaderboard. In our analysis of the benchmarking results we have shown substantial cross-lingual transfer between the Mainland Scandinavian languages, and no notable transfer between the group of Mainland Scandinavian languages and the group of Insular Scandinavian languages. This is the justification for including only the Mainland Scandinavian languages in the online leaderboard while maintaining support for the Insular Scandinavian languages in the scandeval package.

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

# A Cross-lingual transfer experiment

| Language | Hugging Face Model ID | # Parameters |
|---|---|---|
| Danish | `vesteinn/DanskBERT` | 124M |
| Swedish | `KB/bert-base-swedish-cased` | 125M |
| Norwegian | `patrickvonplaten/norwegian-roberta-base` | 125M |
| Icelandic | `mideind/IceBERT` | 124M |
| English | `roberta-base` | 125M |
| German | `deepset/gbert-base` | 110M |
| Dutch | `pdelobelle/robbert-v2-dutch-base` | 117M |
| Finnish | `TurkuNLP/bert-base-finnish-cased-v1` | 125M |
| Russian | `DeepPavlov/rubert-base-cased` | 178M |
| Arabic | `asafaya/bert-base-arabic` | 111M |

Table 3: The Hugging Face Hub model IDs of the models used in the cross-lingual transfer experiment.

| Model | Danish Score | Norwegian Score | Swedish Score | Icelandic Score | Faroese Score |
|---|---|---|---|---|---|
| Danish | **63.87 ± 1.26** | 53.74 ± 3.73 | 52.08 ± 2.70 | 30.39 ± 1.55 | 45.26 ± 1.35 |
| Norwegian | 46.30 ± 2.83 | **58.78 ± 1.44** | 46.90 ± 2.79 | 28.85 ± 1.45 | 43.35 ± 2.20 |
| Swedish | 45.81 ± 2.96 | 47.32 ± 2.66 | **69.29 ± 1.40** | 28.69 ± 1.61 | 43.63 ± 1.90 |
| Icelandic | 30.20 ± 1.23 | 28.68 ± 2.91 | 36.80 ± 2.14 | **71.00 ± 1.50** | **48.26 ± 4.76** |
| Finnish | 32.55 ± 1.47 | 30.71 ± 2.14 | 38.94 ± 1.51 | 16.33 ± 1.89 | 36.87 ± 1.17 |
| English | 34.11 ± 2.11 | 30.92 ± 2.69 | 39.24 ± 1.92 | 28.39 ± 2.41 | 40.75 ± 1.59 |
| German | 28.13 ± 2.04 | 27.58 ± 2.90 | 37.62 ± 4.18 | 26.13 ± 1.63 | 41.02 ± 1.46 |
| Dutch | 31.78 ± 1.62 | 28.27 ± 2.51 | 35.06 ± 1.87 | 26.21 ± 1.79 | 40.83 ± 1.70 |
| Russian | 33.91 ± 1.88 | 33.55 ± 2.17 | 39.14 ± 2.33 | 29.96 ± 1.58 | 43.17 ± 1.66 |
| Arabic | 22.89 ± 1.82 | 19.98 ± 2.24 | 25.40 ± 2.69 | 10.33 ± 2.19 | 35.33 ± 1.57 |

Table 4: The raw benchmarking results used in the cross-lingual transfer experiment.

# B  Training Data Size Experiment

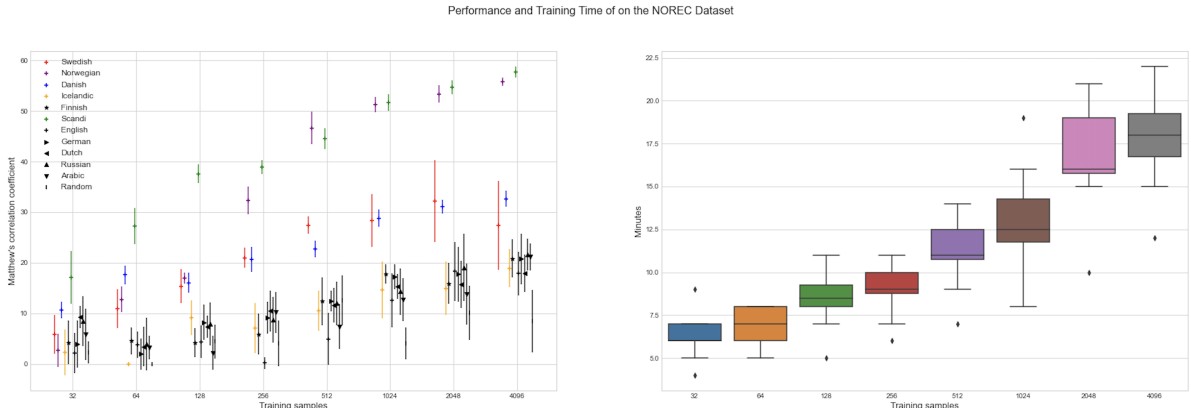

Figure 3: The results from the training data size experiment for the `NoReC` dataset.

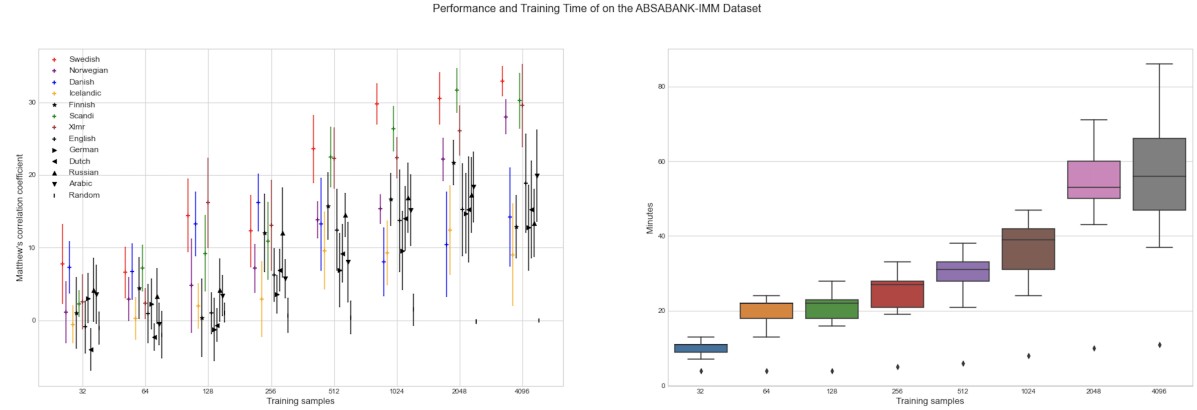

Figure 4: The results from the training data size experiment for the `Absabank-Imm` dataset (Adesam et al., 2020).

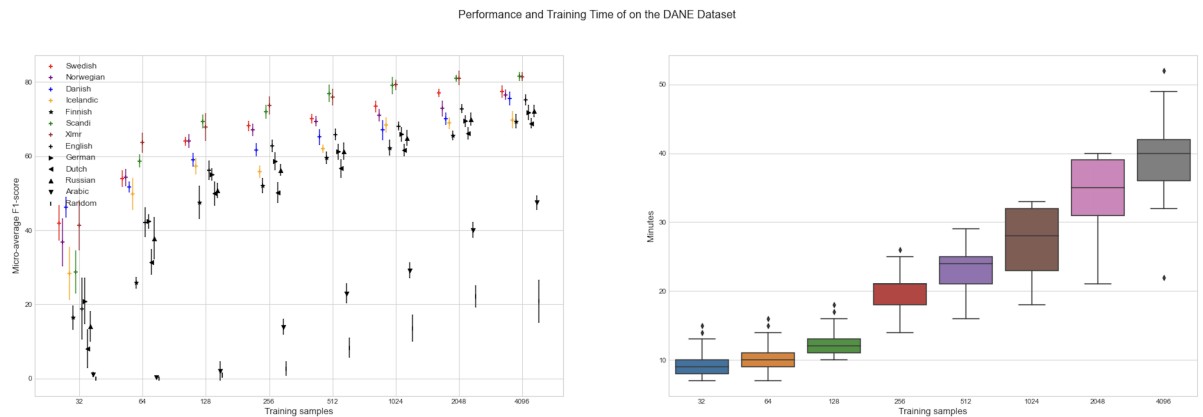

Figure 5: The results from the training data size experiment for the `DaNE` dataset.

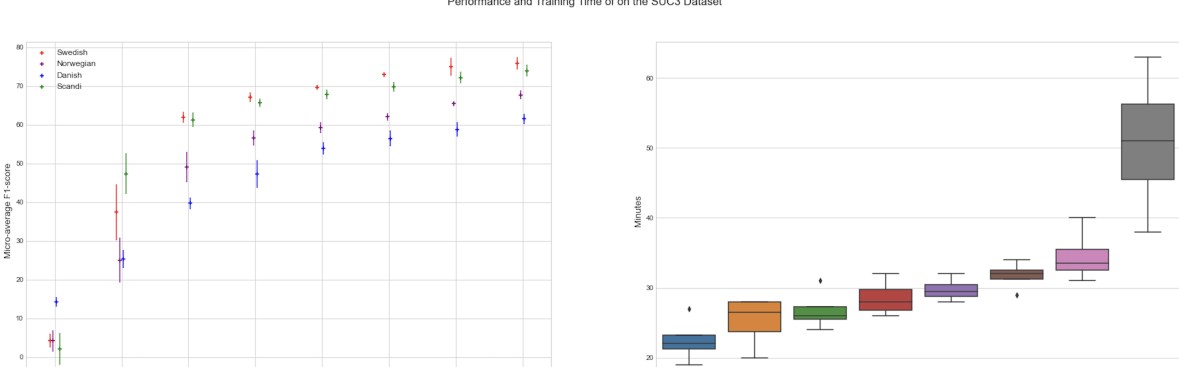

Figure 6: The results from the training data size experiment for the `SUC3` dataset.

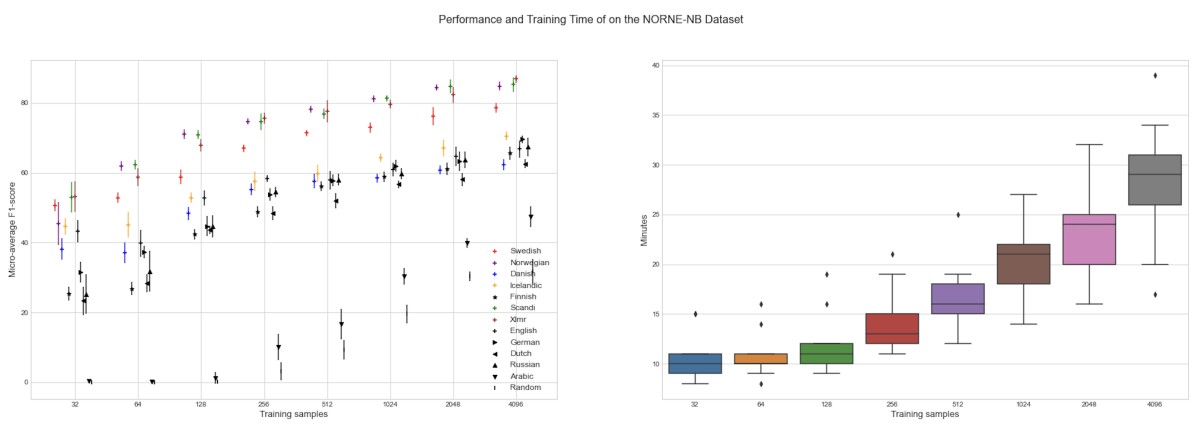

Figure 7: The results from the training data size experiment for the `NorNE-NB` dataset.

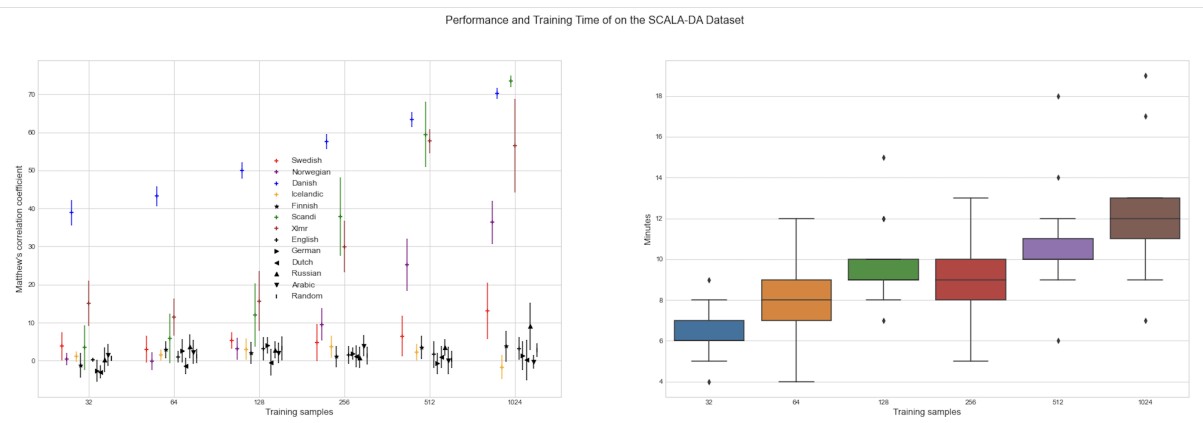

Figure 8: The results from the training data size experiment for the `ScaLA-DA` dataset.

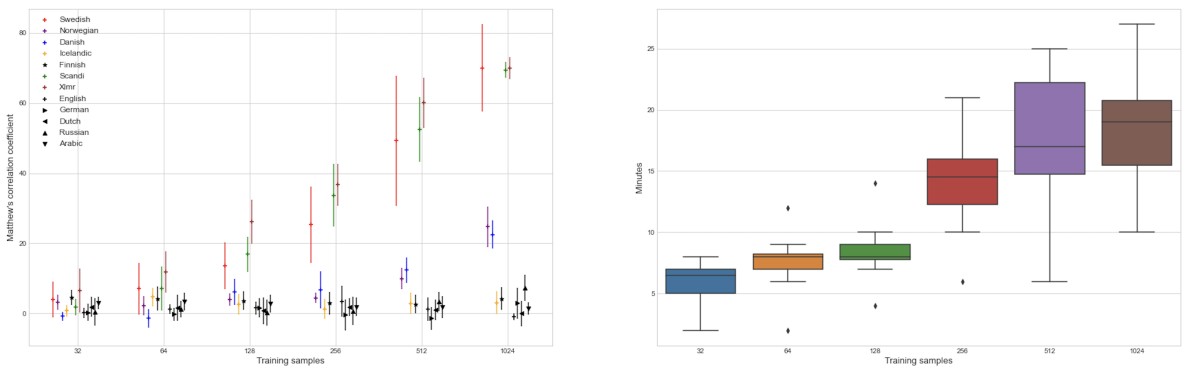

Figure 9: The results from the training data size experiment for the `ScaLA-SV` dataset.

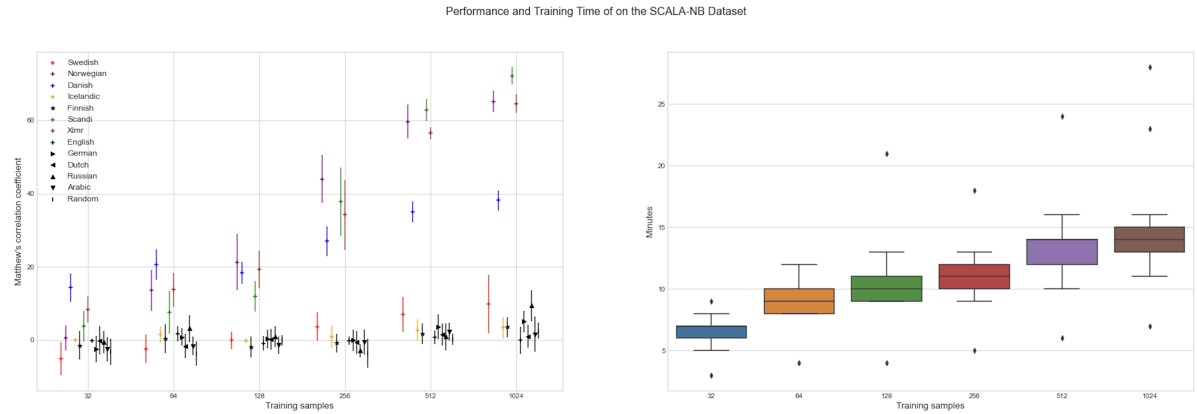

Figure 10: The results from the training data size experiment for the `ScaLA-NB` dataset.

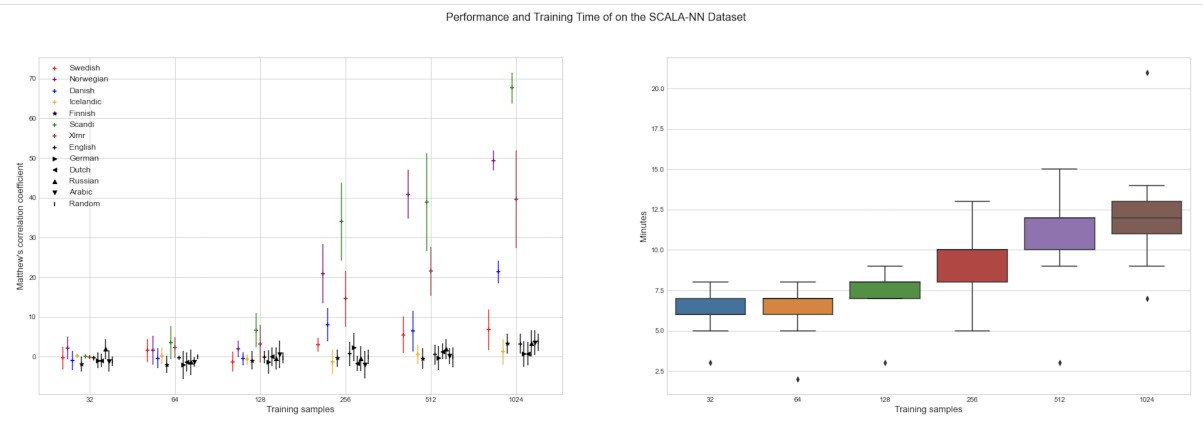

Figure 11: The results from the training data size experiment for the `ScaLA-NN` dataset.