# OpenReview forum: "ScandEval: A Benchmark for Scandinavian Natural Language Processing"
_NoDaLiDa/2023/Conference — NoDaLiDa 2023_

### Official Review · Reviewer_BEm2 · 2023-03-04
**Benchmarking Large Language Models for Danish, Norwegian, Swedish**

**Rating:** 9
**Confidence:** 3

**Review:**

This paper describes datasets (incl. 2 new datasets) and evaluation methods for 4 tasks to benchmark large language models for five Scandinavian languages. This is a worthwhile effort. The benchmarking will set a new standard for the languages involved.

The paper is well-structured and describes the approach in detail from a technical perspective (including finetuning parameters and inference speed). Unfortunately, the paper does not give a single language example for any of the 4 tasks (that are described in section 4). I suggest to add some examples to make the paper more concrete and lively.

Puzzling
- Why does the Danish model have substantially lower performance? (page 7)

The paper has a few wording errors and other stylistic shortcomings:
- in the abstract: the the --> the
- page 1: to choose the correct model --> to choose the best model
- Faarlund (2019) even argue --> Faarlund (2019) even argues
- page 4: corresponding to proper names --> corresponding to person names
- we replace all but the first token in each word --> ??? unclear! ???
- page 5: to ensure that the resulting sentence is indeed correct --> to ensure that the resulting sentence is indeed grammatically incorrect
- page 7: results presented Section 6.1 --> results presented in Section 6.1
- page 8: the left part of Figure 1 is unreadably small
- page 8: at a national level --> at a large language-specific level

There are a number of incomplete entries in the reference list. For example.
Adesam et al. (2020)
Malmsten et al. (2020)
Plaut et al. (1986)
Please specify the publication source (conference or journal or ...) for each entry.

**Paper Type:**

Long paper

---

### Official Review · Reviewer_LRfd · 2023-03-09
**small but robust benchmark for under-represented languages**

**Rating:** 8
**Confidence:** 5

**Review:**

This paper presents a benchmark made of four datasets for diverse NLP tasks in Danish, Norwegian and Swedish. The idea is similar to GLUE and related benchmarks, i.e. testing LLMs by fine-tuning them on specific datasets. This idea is executed well in this work, which brings a good level of automation in the picture. A software library is provided that provides functions to download and manage the data. Furthermore, the transformers library is integrated, making it easy to benchmark LLM that are in the Huggingface repository.

It is less clear if it is possible (and how it it would be) to play around with the hyperparameters, network structure, of even if it is possible to apply the benchmarks to different solutions, e.g. prompt-based learning or non-NN models. Given that the source code is available, and the library is well documented, these options are probably all possible, just not very well discussed in the paper.

The paper is complemented with experiments, including cross-lingual tranfer tests, which showcase the flexibility and utility of the benchmark.

**Paper Type:**

Demo

---

### Official Review · Reviewer_wx8A · 2023-03-10
**Paper on a very important topic of LLM evaluation for Scandinavian languages, but not without its flaws.**

**Rating:** 7
**Confidence:** 4

**Review:**

The paper presents Scandeval, which is a set of NLP benchmarks for Scandinavian languages, accompanied with a public leaderboard and a Python package. I believe this is an important contribution to Scandinavian NLP, but there are some notable issues with the paper:

1) The authors use "a new question answering dataset for the Mainland Scandinavian languages". I have serious doubts about this dataset, due to machine translation and learned semantic similarity estimations involved in its creation. This is acknowledged by the authors themselves (section 4.4), and to my mind, it seriously undermines using this dataset to evaluate any models, especially on a large scale. May be it's better to wait until proper Q&A datasets appear for Scandinavian languages, since the results on this dataset can be misleading.

2)  Another benchmark introduced by the authors is "a new linguistic acceptability dataset for all the Scandinavian languages, dubbed ScaLA". The description of the creation  of this dataset in section 4.3 is very shallow. At the very least, I would expect some statistics (we don't even know how large the dataset is). But what is even more important, I am not sure the described workflow actually yields a correct linguistics acceptability dataset:
- How can we be sure that all the sentences in UD treebanks are grammatically correct? These corpora might contain mistakes and garbled sentences; UD even has special tags for incorrect forms.
- The heuristics used to make the sentences non-grammatical look sane by themselves, but they are not evaluated in any way. How can we be sure that they do not produce correct sentences in, say, 50% of the cases?
In the end, again, the results on ScaLA can be misleading, I belive.

3) What does "bootstrapped test sets" exactly mean (section 3.3)?

4) Any explanation of why Matthews correlation is used as the primary metrics for sentiment classification, and not macro-F1 score?

5) Regarding the inference speed measures in section 4.5. Although it's great that inference speed finally gets the attention it deserves, I am a bit surprised that the author uses a dummy document for it. Why not employ some sort of a sample from your test sets, yielding much more realistic input?

6) Figure 1 is difficult to parse, with the font size being too small. In general, section 6.3 (which this plot illustrates) could benefit from some clear explanation of what hypothesis it aims to verify or falsify, and what is the conclusion for this section.

7) You probably don't need that many footnotes with the Scandeval URL (anonymized or not). One is enough.

IMPORTANT!!!
The paper is actually not anonymous: line 160 points at the HuggingFace space with the datasets, and it has only one owner with clearly stated name and affiliation. Not sure how to deal with it, leaving this to the conference organizers.
My personal opinion is that papers describing such projects (leaderboards, benchmarks, web services, etc) are always difficult to anonymize. So probably we just have to live with it.

**Paper Type:**

Long paper

---

### Decision · Program_Chairs · 2023-03-17

Accept